# Could Selenium Supplementation Prevent COVID-19? A Comprehensive Review of Available Studies

**DOI:** 10.3390/molecules28104130

**Published:** 2023-05-16

**Authors:** Roberto Ambra, Sahara Melloni, Eugenia Venneria

**Affiliations:** Consiglio per la ricerca in agricoltura e l’analisi dell’economia agraria (CREA)—Research Centre for Food and Nutrition, Via Ardeatina 546, 00178 Rome, Italy; sahara.melloni@crea.gov.it (S.M.); eugenia.venneria@crea.gov.it (E.V.)

**Keywords:** selenium, supplementation, COVID-19, prevention

## Abstract

The purpose of this review is to systematically examine the scientific evidence investigating selenium’s relationship with COVID-19, aiming to support, or refute, the growing hypothesis that supplementation could prevent COVID-19 etiopathogenesis. In fact, immediately after the beginning of the COVID-19 pandemic, several speculative reviews suggested that selenium supplementation in the general population could act as a silver bullet to limit or even prevent the disease. Instead, a deep reading of the scientific reports on selenium and COVID-19 that are available to date supports neither the specific role of selenium in COVID-19 severity, nor the role of its supplementation in the prevention disease onset, nor its etiology.

## 1. Introduction

Selenium is an essential trace element with important antioxidant properties, and anti-inflammatory and immune functions [1]. Selenium modulates the biochemical activity of several enzymes, acting directly as their redox center, and has pharmacological effects, including antimicrobial, pro-oxidant, pro-apoptotic, anticancer and toxic, depending on the concentration and the type of derivative compound used [2,3]. Several papers have suggested that selenium supplementation could be used to ameliorate biological functions; however, even if randomized trials have indicated that selenium supplementation has a beneficial effect, metanalyses have shown that the results are inconclusive [4,5,6,7]. In 2009, the European Food Safety Authority (EFSA) proposed that selenium is indispensable for the oaptimal function of the immune system [8]; however, only adequate intakes have been established [9], and currently supplementation is not recommended for disease prevention. Similarly, clinical observational studies on the association between low selenium in serum or plasma and all-cause mortality are contradictory, showing a higher risk in prospective studies in the US and Europe [10,11,12,13] but not in China [14]; this shows the high level of geographical variation, ranging from toxic to deficient intakes [15]. Notably, more recent work in China has indicated that selenium intake is inversely associated with all-cause mortality [16].

Because of the urgent need for solutions to cope with the SARS-CoV-2 emergency, starting immediately after the onset of the COVID-19 pandemic, a huge number of publications focusing on the relationships between nutrients and COVID-19 were produced, mostly speculative. For the first year of the pandemic, Scopus.com returns for the keywords “COVID-19” and “selenium” included 41 reviews, letters, editorials or books out of 73 total publications. On the other hand, only a few scientific works are currently available on the role of selenium and its supplementation in COVID-19 etiology. This work analyzes the accuracy and reliability of the information provided by scientific reports regarding the relationship between selenium status and/or supplementation and COVID-19 etiopathogenesis. Moreover, we have analyzed the validity of the information reported by some highly cited reviews on selenium and COVID-19 relationships, particularly those based on coxsackievirus B3 or influenza virus mice fed with selenium-deficient diets [17].

## 2. Methodology

The search carried out in this review was performed, as described in Figure 1, on December 2022. In order to identify the selenium-specific literature on COVID-19, title/abstract searches were conducted in the Scopus database, using only the terms “selenium” and “covid”, with no date limitation. Nonduplicate records retrieved from Scopus.com (*n* = 364) were deeply read for their content by three independent researchers in nutrition, finding 97 non-pertinent publications (68 scientific articles, 23 reviews and 6 letters) that dealt with, among other arguments, antiviral drugs (21), the nutritional status during the pandemic (14), the detection of SARS-CoV-2 (8) and air pollution during the pandemic (5). Among the remaining 267 pertinent publications that dealt with selenium and COVID-19, there were 6 editorial letters, 211 reviews (including 88 exclusively speculative ones) and 50 scientific studies. Nine studies were further excluded: three were clinical supplementation studies that administered selenium together with other ingredients [18,19] or drugs [20]; one was a cross-sectional analysis that actually did not include COVID-19 patients [21]; three included only one patient [22,23,24]; and two were not available in English [25,26].

Of the remaining 41 pertinent scientific articles, only 2 were clinical supplementation studies, 26 were clinical observational studies, 10 were ecological and 3 were genetic (Figure 2). In total, 15 studies concluded that there was an absence of a significant association, while 26 publications claimed that selenium has some kind of association with COVID-19 or has a protective role. As none of these reports bring, in our opinion, sufficient supporting data to conclude that selenium has an effect against COVID-19, their analysis is reported separately, after those concluding that there is an absence of association.

## 3. Results

### 3.1. Publications Reporting No Association between Selenium and COVID-19

Among the 15 publications concluding that there is an absence of association between selenium and COVID-19, 5 were ecological studies (Table 1), 7 were clinical observational studies (Table 2) and 3 were studies associating genetically predicted nutrients levels with COVID-19 outcomes. No clinical supplementation studies were found.

#### 3.1.1. Ecological Studies (Table 1)

The Spanish group of Prof. Francisca Serra submitted, in July 2020, a “current state of evidence” review dealing with the nutritional and nutrigenetic factors that affect immunity; it included a compilation of COVID-19 epidemiological data that were available at the time (May 2020) in 10 European countries, showing a non-significant inverse correlation between the suboptimal estimated status of selenium and COVID-19 incidence [27]. Later, the same group focused on the estimated selenium intake in Spain and did not find any significant association with the epidemiological indicators of COVID-19 [28].

In their review entitled “Biological Role of Trace Elements and Viral Pathologies”, Ermakov and Jovanović analyzed the association between COVID-19 incidence in various areas of Russia (on 5 September 2020 and on 29 January 2021) and selenium status, scored using previous geographical data of blood serum selenium (2017) or previous geographical data (2001) of environmental selenium (herbaceous plants, surface and groundwaters, and annual precipitation), and white muscle disease incidence in farm livestock [29]. Even if some negative correlation was present between environmental selenium, no correlation was found with the geographical data of blood serum selenium, possibly because of the dietary compensation of the microelement [29].

On the other hand, Chen and coworkers studied the geographical relationship between soil trace elements and the COVID-19 fatality rate at the county level in the USA, using data from the National Geochemical Survey and from the Johns Hopkins University (erroneously cited as “John Hopkinson”) [30]. Unfortunately, as for other ecological studies (see below Section 3.2.2), the authors used dated geochemical concentrations of trace elements (from 1997 to 2009). With respect to the COVID-19 fatality rate, seven time points were used from 8 October 2020 to 25 March 2021. The authors argued for some association between the fatality rate and the selenium soil concentration in the early period (up to 3 December 2020), but no robust association was found following 31 December 2020. Looking at their data, it appears that, starting from this date, the highest quartile of soil selenium concentration increased the COVID-19 fatality rate. However, as the details of the statistical analysis are not shown, no conclusions can be made.

Finally, in a cross-sectional study performed using a face-validated questionnaire-based study submitted in Jordan between March and July 2021, in 2148 non-vaccinated recovered COVID-19 adult individuals, no association was found between the COVID-19 disease severity and the use of selenium supplementation before COVID-19 infection, after adjustment for the confounders [31]. Table 1 summarizes the ecological studies.

#### 3.1.2. Clinical Observational Studies (Table 2)

All 7 clinical observational studies (Table 2) have important limitations, including a small sample size, a lack of dynamic analyses throughout the hospitalization period and a poor quantification of supplement intake before and during hospitalization.

One study only looked retrospectively at the mean serum selenium level of 33 COVID-19 patients (out of 72 with mean age 57.1 ± 9.8 years) at Intensive Care Unit (ICU) admission (West London, UK, March–May 2020), finding it to be within the normal range despite a 33% mortality [32]. Voelkle and coworkers concluded that, even though selenium deficiency was found in 51% of 57 COVID-19 cases consecutively admitted between 17 March 2020 and 30 April 2020 at the Cantonal Hospital Aarau (Switzerland), “no association between low selenium levels and adverse clinical outcomes was found” [33]. No prevalence of hyposelenemia was observed in 152 COVID-19 patients compared to 88 non-COVID-19 patients that were admitted to the COVID-19 screening centers of the University Hospitals of Nancy and Marseille (France) from 24 April to 23 May 2020 [34]. Karakaya Molla et al. enrolled 49 COVID-19 pediatric patients (mean age 13 ± 3) admitted at the Ankara Pediatric hospital (Turkey) between 15 May and 15 June 2020, finding normal selenium levels [35]. Tayyem and coworkers used a bigger sample of 367 hospitalized COVID-19 patients in Prince Hamza Hospital (Amman, Jordan) from 17 March and 25 July 2020, and found no association between selenium and disease severity (mild, moderate, severe and fatal) [36]. However, as stated by the authors themselves, the estimation of the selenium intake from hospital food was inacurate due to uncontrolled food consumption outside the hospital. Bagher Pour and coworkers compared COVID-19 severity and its outcomes in ICU (*n* = 114) and non-ICU (*n* = 112) patients admitted to the Reza Hospital of Tabriz (Iran) from 10 October to 10 December 2020 [37], finding no significant differences in the serum selenium levels between the two groups (unexpectedly, they found a positive trend in ICU patients), nor when comparing those that died (*n* = 56) with those that were discharged (170) before the end of the study. Notably, differences were found in the serum zinc levels, i.e., low zinc levels were associated with death among COVID-19 patients. Finally, Razeghi Jahromi et al. reported that following adjustment for confounding factors, the negative association between the serum selenium level and COVID-19 severity (mild, moderate and severe) that was found in 84 patients diagnosed with COVID-19 at admission (up to 1 September 2020) to Sina Hospital (Tehran, Iran) lost its significancy, bringing the authors themselves to the conclusion that “the effect of confounding factors especially age are stronger than the effect of serum selenium level in predicting COVID-19 severity” [38].

#### 3.1.3. Genetic Studies

Kotur and coworkers genotyped 73 adults, hospitalized (between April and June 2020) at the Belgrade University Clinical Centre with mild (*n* = 35), moderate (*n* = 21) and severe (*n* = 17) COVID-19, for variants of the Dimethylglycine dehydrogenase gene (rs17823744) [39], and found that homozygous carriers of the A allele, previously associated with lower levels of serum selenium [40] and a greater increase in serum selenium following supplementation [41], had a lower incidence in patients with severe COVID-19 [39]. However, as stated by the authors themselves, after correction for age and gender, the association lost its statistical significance, possibly because of the small size of the cohort and an inhomogeneous age distribution between the groups. Accordingly, no significant association between genetic variants and an increased risk of symptomatic COVID-19 could be found in pediatric COVID-19 patients categorized into two groups, i.e., asymptomatic (enrolled, *n* = 20) and symptomatic (hospitalized, *n* = 22) [41]. Unfortunately, direct measurements of the selenium status were not performed in neither of the two subgroup studies.

More recently, two Mendelian Randomization (MR) analyses of genetically determined selenium levels were published [42,43]. In the first one, Sobczyk and Gaunt merged the outcomes (infection, hospitalization and critical) of 49,562 COVID-19 patients (and 2 million controls, from 35 European-only cohorts) with the selenium-level-predicting genetic variants obtained in previous genome-wide association studies [42]. The authors stated that there was “a weak causal effect of meta-analyzed selenium levels on SARS-CoV-2 infection, COVID-19 hospitalization and critical illness” and concluded that “the utility of dietary supplementation for general population in the COVID-19 pandemic may be limited”. In the second study, Daniel et al. replicated the findings of Sobczyk and Gaunt using the same MR approach but with larger sample size (87,870 individuals of European descent with a COVID-19 diagnosis and 2 million controls); this study had a higher statistical power and better robustness analysis, concluding that there was poor evidence of a possible association between selenium and COVID-19 outcomes [43].

### 3.2. Publications Presenting Inconsistent Supporting Data

Among the 26 studies, 2 were clinical supplementation studies, 5 were ecological studies and 19 were clinical observational studies. No genetic studies claiming that there is an association were found.

#### 3.2.1. Clinical Supplementation Studies (Table 3)

As mentioned above, only two clinical supplementation studies were found, which are summarized in Table 3. The first one is assimilable to an intervention study as it was carried out on 72 COVID-19 ICU-hospitalized patients with acute respiratory distress syndrome (ARDS) that were treated with different therapeutic regimens and supplements for 14 days, from 22 February 2020 [44]. Jamaati and coworkers “recommend the administration of selenium in the ARDS patients” [44]. However, this recommendation cannot be attributed to COVID-19-associated ARDS nor to ARDS alone, as all patients received the same treatment of supplements, i.e., the study lacked a control group. In the second one, Notz and coworkers retrospectively analyzed the serum selenium levels of COVID-19 critical patients supplemented with nutrients at the University Hospital of Wuerzburg between 20 March to 31 October 2020 [45]. They reported that supplementation was associated with higher levels of serum selenium in those surviving ICU (10 to 14 days), compared to those who died during ICU (*p* = 0.075), and concluded that adequate selenium (and zinc) supply “may potentially be of clinical significance for an adequate immune response in critically ill patients with severe COVID-19 ARDS” [45]. The work has several biases and limitations. Firstly, as stated by the authors themselves, the “finding might be biased by small n-numbers”. In fact, data were available for only 16 out of 72 patients. Moreover, nutrient supplementation also contained zinc and was not identical for all patients, as different combinations of artificial nutrition were used apparently randomly. Because of its retrospective design, no differentiation of supplementation based on a normal vs. low selenium status at admission was performed, nor was a control group without supplementation available. Finally, no control group of critically ill patients with the same characteristics but without COVID-19 was used, deeply questioning the SARS-CoV-2-related specificity of the findings.

**Table 3 molecules-28-04130-t003:** Supplementation studies reporting association between selenium and COVID-19.

First Author, Year [Ref]	Country	Study Period	N of Subjects Involved	Study Population (Sex, Mean Age)	Selenium Plasma Levels	Supplementation	Follow Up
Jamaati, 2020 [44]	Iran	Two weeks, from 22 February 2020	232	No data	No data	200 µg/day	No
Notz, 2021 [45]	Germany	20 March–31 October 2020	22	Males *n* = 14, 36%, females *n* = 8; mean age 60.5 years (50–69)	No data	1 mg of Se per day	Yes

#### 3.2.2. Ecological Studies (Table 4)

Chronologically, the first statement of an ecological association comes from a Letter to the Editor reporting a population-based analysis of website-reported (www.baidu.com, accessed on 14–18 February 2020) COVID-19 incidence, recovery and mortality rates (only provinces or municipalities with >200 cases and cities with >40 cases) from 14 to 18 February 2020; this analysis was performed in Chinese regions with huge differences in their intake and hair concentrations of selenium [46]. We found different biases. Firstly, as indicated by the authors themselves, the data on the regions’ selenium intake and hair concentrations were dated, mostly dating from 2011, and several confounders were not evaluated (age, chronic diseases and therapies). Secondly, the authors stated that the selenium status was available only for two cities inside the Hubei Province; however, selenium status is mentioned only for the city of Enshi. Moreover, even if the city of Enshi (death rate 1.6) is reported for being an exception to the high mortality rate of the Hubei Province (high death rate), based on the data shown, other cities (Shiyan and Xianning) inside the Hubei province have a lower death rate (0.3 and 1.2). Notably, a higher incidence in Shiyan and Xianning, compared to that of Enshi, was reported later by others (using the incidence of diagnosed people by April 30, when the city of Wuhan was unsealed) even if it was not, unexpectedly, associated with a higher mortality [47]; this indicates that some other factor, rather than selenium, was involved in the observed differences in COVID-19 lethality between these cities. Accordingly, looking more carefully at the two reports, it appears that other cities had a much greater incidence than Enshi (even 20 times higher than in Ezhou) without a proportional increase in mortality [46,47]. The same group of authors also later claimed that there was a significant association between COVID-19 death rates in 46 US states (assessed on 27 August 2020) and atmospheric dimethyldiselenide, which was extrapolated from the selenium concentration in alfalfa leaves [48]. As for the other ecological studies, the categorization of the selenium concentration is poorly reliable, being from 1997.

Another regional association study between COVID-19 case fatality rate and selenium deficiency was reported by Zhang et al. [49]. The authors extrapolated selenium deficiency from Chinese local levels of selenium in crops and topsoil, and subdivided this into three classes (non-deficient, moderate and severe). Compared to the above work by Zhang et al. [46], more recent data (8 December–13 December 2020) and a higher number of cases (14,045) and cities (147) were included. However, in order to avoid biases due to medical and health shortages, the province of Hubei was excluded, being the most affected. Several important biases are present. Again, “the nationwide data on selenium concentration in crops or topsoil are old”, especially for crops (1985). Moreover, the authors’ hypothesis that lockdown conditions may have reduced the use of non-local crops, thanks to national data on grain and vegetable self-sufficiency, is not supported by consumption nor sales data; their work is also affected by the known migration bias of ecological studies [50]. Lastly, only data for the 10 cities with the highest Case Fatality Rates (CFRs) were provided, and cities without deaths or with the lowest CFRs were not included in the analysis. In conclusion, especially compared to other countries, the number of deaths is very limited and could explain the lack of effect on COVID-19 CFR by demographic, social or medical variables.

A population-level study included data from 170 countries and performed a quantitative cross-sectional analysis of 2017′s FAO data of food supply (expressed as food items and their nutritional contents of macronutrients) vs. the proportion of individuals who recovered or died from COVID-19 until January 2021 (from JHU-CSSE) [51]. The authors concluded that foods with a higher content of selenium had a positive effect on COVID-19 recovery in developed countries, but not in in countries with extreme poverty [51]. Again, an important limitation of the work is that, because of the pandemic, 2017′s food supply data may not accurately represent the years 2020–2021. Specifically, the authors’ statement that no significant changes in food supply occurred worldwide in the past four years cannot be applied to the pandemic because of its evident effects on logistics and eating habits. Moreover, the positive association between selenium and patients’ recovery from COVID-19 was indirectly inferred from the higher consumption of virtually any type of food item (milk, meat, fish, seafood, eggs, vegetables) containing any possible micronutrient; thus, this association is not unambiguously attributable to the intake of selenium. Accordingly, as stated by the authors, the positive effect on COVID-19 recovery was not observed in countries with a higher global hunger index [51].

The last ecological study available was based on the evaluation of selenium (and other nutrients) intake, from 138 food items, by means of a Food Frequency Questionnaire (FFQ) that was submitted to 500 WhatsApp-recruited Iranians with a history of COVID-19 diagnosis but not treated in hospital, and to as many non-COVID-19 Iranians [52]. The authors reported a significantly lower selenium intake in COVID-19 individuals and claim that “an increase of one unite in selenium consumption reduced the risk of COVID-19 incidence by 91%”. As stated by the authors themselves, the work has the important limitation of being based on a non-face validated electronic questionnaire, from which the consumption frequency of each food was calculated by considering the amounts consumed during the previous year. Other shortcomings included the fact that the confidence intervals were incorrect, as they did not include the reported value, and the lack of adjustment for confounders. The study also lacked the verification of SARS-CoV-2 negativity for non-COVID-19 individuals and their vaccination status (the study was submitted for publication in July 2021). Table 4 summarizes the ecological studies.

**Table 4 molecules-28-04130-t004:** Ecological studies reporting association between selenium and COVID-19.

First Author, Year [Ref]	Country	Study Period	Consulted Databases for Selenium Levels	Consulted Databases for COVID-19 Incidence
Zhang, 2020 [46]	China	14–18 February 2020	Crop selenium content from the Chinese Academy of Agricultural Sciences	COVID-19 mortality data National Bureau of Statistics of China (www.stats.gov.cn, accessed on 14–18 February 2020)
Zhang, 2022 [48]	China, USA, UK	27 August 2020	Selenium concentration in land areas from the US Census summary 2010 and Bureau 2012	US COVID-19 case number and case mortality from CDC (https://www.cdc.gov/coronavirus/2019-ncov/COVID-data/previouscases.html, accessed on 27 August 2020)
Zhang, 2021 [49]	China, United States	8–13 December 2020	Hair and soil selenium concentration in Hubei provinces from the literature	Confirmed COVID-19 cases collected by realtime data from the Baidu website, a non-governmental website
Cobre, 2021 [51]	Brazil	27 January 2021	Food supply and COVID-19 recovery data from Kaggle Platform (US government)	
Mohajeri, 2022 [52]	Iran	Not specified, submission date July 2021	Food intake data (converted to antioxidants intake: vitamin A, vitamin C, vitamin E, vitamin D, zinc, selenium) collected using food frequency questionnaires	COVID 19 status collected by electronic questionnaires

#### 3.2.3. Clinical Observational Studies (Sorted Chronologically by Execution Period, Table 5)

The earliest clinical observational study correlating selenium levels with COVID-19 severity was performed at the Qilu Hospital of Shandong University (China) between 27 January and 18 May 2020 [53]. The authors concluded that “micronutrients should be considered in the treatment of COVID-19, especially in critically ill patients”. However, the study lacked controls and involved only 16 patients (8 mild and 8 severe) with different ages and BMIs (statistically different between the two COVID-19 severity groups) [53].

Im and coworkers compared the nutritional status of 50 COVID-19 patients hospitalized at the Inha University Hospital (South Korea) from February to June 2020 with that of 150 healthy age/sex-matched controls [54]. They concluded that the “results suggest that a deficiency of selenium may decrease the immune defenses against COVID-19 and cause progression to severe disease”. One important bias of their work is that the selenium samples were inhomogeneous and incomplete (“within 7 days of admission”), i.e., possibly at different stages of the disease. Moreover, no correlation analysis was performed between the level of selenium deficiency and the disease severity.

Skalny et al. correlated the clinical parameters of the illness severity of 150 COVID-19 patients (three groups of *n* = 50 each, mild, moderate and severe, plus 44 control healthy subjects) with their serum selenium at hospital admission (Sechenov University Clinical center, Moscow, Russia, between March and June 2020) [55]. They concluded that the reduction in serum selenium was inversely correlated with lung damage. However, the age of the severe group did not match that of other groups, being, on average, 10 years higher. Moreover, looking at the figures, statistical validity does not seem likely because of the huge standard deviation of the serum selenium.

Pincemail and coworkers observed, in a pilot study performed in May 2020 at the University Hospital of Liège (Belgium), that COVID-19 patients staying for less time in the ICU (7–11 days) had, unexpectedly, lower selenium levels compared to long-term ICU stayers (38.7–43 days) [56]. They concluded that selenium deficit contributes to alterations in the systemic oxidative stress of critically ill COVID-19 patients. Unfortunately, the study involved only nine COVID-19 patients (three short- and six long-term ICU stayers), for which several data are lacking, i.e., individuals’ selenium status at admission and co-morbidities. Moreover, because of extreme variations in the individual nutritional support, only the mean intake of nutrients was considered, without distinguishing between short- and long-term stayers.

One study analyzed (retrospectively) the association between urinary selenium levels and the COVID-19 severity or outcome in 138 patients hospitalized at the Tongji Hospital of Wuhan, Hubei (China), presumably within June 2020 (this is the submission date of the manuscript) [57]. The authors concluded that COVID-19 severity, particularly with regard to liver dysfunction, correlates with increased urinary levels of selenium. However, the data on urine samples were incomplete and no indication of the time point of urine sampling was given. This is particularly relevant because the longitudinal analysis shows that the main oscillations in the selenium levels occurred within the first weeks after the onset of symptoms, but the durations of the hospital stay (8–32 days) and disease onset (11–41 days) were very different.

Al-Saleh et al. reported a 18% decrease in selenium levels in patients with severe COVID-19 symptoms compared to those with lower ones [58]. The study comprised 155 COVID-19 patients hospitalized between June and July 2020 at the King Faisal Specialist Hospital and Research Center (Riyadh, Saudi Arabia) that were grouped based on disease severity (asymptomatic, mild, moderate and severe). The authors concluded that there was an independent association with COVID-19 severity based on the observation that the decrease in selenium was higher (going up from 18 to 30%) following adjustment for inflammatory markers. However, as “99 patients took supplements including vitamin D (37) and minerals (15)”, no conclusions can be made regarding the potential role of selenium supplementation in COVID-19 severity or outcome.

The group of Prof. Schomburg published three papers dealing with selenium status and COVID-19 survival rates. In the first one, they consecutively sampled a total of 166 serum selenium samples from 33 COVID-19 patients hospitalized at the Public Hospital Klinikum Aschaffenburg-Alzenau (Germany) for 10 days (dead patients, *n* = 6) or 19 days (discharged ones, *n* = 27) within July 2020 (the date of manuscript submission) [59]. Because of an inverse correlation between selenium levels and COVID-19 survival rates, they concluded “that selenium status [selenium and selenoprotein P] analysis in COVID patients provides diagnostic information”. However, the data were reported in an aggregated way (differently from zinc) for population groups (dead or surviving) without specifying the use of nutritional supplements. In the second publication (submitted in August 2020), the authors included five more serum selenium samples and two more discharged male patients, and concluded that a normal zinc status and selenoprotein P (SELENOP) status (but not selenium levels) are indicators of high survival odds in COVID-19 and that the supplementation of selenium may support convalescence in patients with a proven selenium deficit [60]. However, as for the first publication [59], no indication of the time points nor of the exact number of blood samples (only average 4.9 ± 4.2 samples per patients) was provided. Such points are crucial as the lengths of the stays were different for the dead and surviving patients. Moreover, a correlation with selenium was found only in surviving patients and not in those who died, which did not show any statistically significant trend (even if authors improperly concluded that “no increase was seen in the non-survivors”). In the third publication (submitted in April 2021), the authors had two more patients (*n* = 35), one less discharged male, and one extra dead female patient; the study used almost five consecutive blood samples per patient (173 instead of 175) [61]. The work has the same limitations and biases, and the methods used for the determination of selenium are completely lacking and references are not even provided. Curiously, the authors disprove the previous indication that there are inverse selenium trends in dead vs. surviving patients during hospital stay, and conclude that “the combined analysis of serum copper and selenium status provides prognostic information on survival odds”, making the entire picture, if possible, even less clear.

In a study involving 61 adult patients admitted to the ICU of the CITIBANAMEX Center (Mexico City, Mexico) between 20 August and 20 September 2020, the selenium concentration was found decreased in severe COVID-19 patients (*n* = 27), compared to the levels found in 25 age- and gender-matched healthy subjects [62]. However, no significant decrease was found in moderate COVID-19 patients (*n* = 34). Inexplicably, the authors excluded the COVID-19 patients that developed moderate or severe pneumonia.

Erol and coworkers compared the serum selenium levels, during gestation, of 71 pregnant SARS-CoV-2-positive women to those of 70 controls with similar and demographic characteristics (between July and December 2020 at the Ankara City Hospital, Turkey); these results were subdivided into the three trimesters of pregnancy, and it was reported that there were significantly lower serum selenium levels in pregnant women with COVID-19 during the second and third trimesters compared to the controls; this is because of the “increased selenium needs depended on the immune response against infection” [63]. However, the controls were apparently not checked for SARS-CoV-2 and the infection course was not reported clearly, i.e., without differentiating groups. Thus, the recommendation that an “adequate level of maternal selenium levels should be taken into consideration along with its potential protective roles in COVID-19 disease” is baseless for the general population.

Majeed et al. analyzed the serum selenium levels of 30 SARS-CoV-2-RT-PCR-positive COVID-19 pauci-symptomatic individuals (age 37.5–43, median 40.5) with fever and dyspnea, but without hypoxemia, that were hospitalized in two Indian hospitals (Apollo in Chennai and Vagus in Bangalore, up to October 2020), versus 30 apparently healthy controls (age 26–37, median 33.5) [64]. The authors reported significantly lower selenium levels and lower oxygen saturation in patients vs. controls, and hypothesized that “supplementation may be useful in reducing the devastation caused by the virus in India”. Since they did not any test supplementation, and because mild/severe COVID-19 was excluded, the hypothesis appears baseless. Moreover, differences were not significant in women, the oxygen saturation level was both a result and separation criterium, the COVID-19 and control groups were different in age, and the SARS-CoV-2 negativity of the latter is missing.

Younesian et al. compared the serum selenium of 50 hospitalized COVID-19 patients admitted to Sayyad Shirazi hospital (Gorgan, Iran) to 50 healthy controls; they concluded that, even if decreased selenium levels were not associated with increased COVID-19 severity and mortality, they may be a risk factor for infection [65]. The work has important shortcomings that undermine its credibility. Details regarding the age and sex distribution of the controls and the COVID-19 severity of the patients are lacking, and even the virus negativity of the controls is not mentioned. Moreover, the authors report in the text that male patients had higher selenium levels than female patients; however, the values in the table, excluded for SD, are identical (77.8 ± 12 vs. 77.8 ± 16.7). Finally, the article was submitted at the end of March 2021, but no information is provided regarding the vaccination status of the patients.

Significantly lower serum selenium levels in COVID-19 inpatients, compared to both COVID-19 outpatients and healthy controls, were reported by Kirankaya et al. [66]. The authors compared the serum selenium levels of 146 children aged 3 to 180 months using COVID-19 PCR diagnosis between December 2020 and May 2021 at the Istanbul Bagcilar Training and Research Hospital (Turkey) to those of 49 healthy controls who had applied to the Pediatric Health and Diseases outpatient clinics for routine controls; the authors conclude that those with lower levels of serum selenium were at a higher risk of COVID-19. However, the SARS-CoV-2 negativity of the healthy controls and the COVID-19 disease severity of the inpatients were not provided, and based upon table data, the controls were younger (70 vs. 120 median months of age).

Kocak and coworkers compared the serum selenium concentrations of 60 COVID-19 patients (divided into four groups based on disease gravity) who had applied to the Infectious Diseases Department of Ataturk University Research Hospital (Turkey) to those of 32 SARS-CoV-2-negative healthy individuals within July 2021 (the date of article submission) [67]. The authors argue that the selenium concentrations were statistically lower (*p* < 0.001) in the patient group. However, the data graphed in Figure 1 is not congruent with the data present in Table 4, questioning the reliability of the results. In fact, the differences in the selenium levels vs. controls were statistically significant only in the severe and mild patient groups, as stated by the authors themselves. Finally, in contrast to what was stated, the demographic characteristics of the controls were different compared to the patients.

Ozdemir et al. analyzed the changes in trace elements in 15 obese COVID-19 moderate patients before and after treatment with Favipavir and Hydroxychloroquine at the Izmir Bakircay University Cigli Training and Research Hospital (Turkey) between January and August 2021, finding that 70% of the patients still had hyposeleniumia [68]. Because the increase in selenium levels after the treatment was statistically significant, the authors claim that “keeping serum selenium levels high in patients via selenium supplementation can lead to an essential breakthrough in the treatment of COVID-19”. Because the authors did not, apparently, supplement patients with selenium, the claim is baseless. Moreover, no information is given regarding the patients’ vaccination status and treatment length.

Du Laing and coworkers reported that the disease severity and mortality rate of 138 COVID-19 patients admitted to two different hospitals (*n* = 79 + 59) in Ghent (Belgium) were associated with low serum selenium levels, especially in patients with co-morbidities [69]. Specifically, the authors claim that the “selenium and SELENOP status showed consistent and relatively linear downward trends with disease severity”. However, based on the presented data, there was not statistical significancy throughout the four severity classes (mild/moderate, severe, critical and death) and the mortality risk was calculated using very limited numbers, i.e., 10 deaths out of 73 patients (instead of 79). The authors claim that there is a correlation between certain co-morbidities and selenium levels. However, the role of age was not considered properly. In fact, based on their data, selenium deficiency appears deleterious only above 65 years; meanwhile, for none of the other risk factors considered (diabetes, tumor, obesity and cardiac disease), selenium status was a determinant for the outcome of death. Thus, the claim that “for the diabetic patients, low Se status at admission was particularly associated with high mortality risk” is misleading. In fact, based on the data presented, the mortality risk was equally high in non-diabetic, non-tumor and non-cardiac disease patients. Only for obesity does patients’ selenium status appear to have a differential role, being, however, a risk determinant only in non-obese patients, which is somehow unexpected. Therefore, the conclusion that “selenium status can thus be considered as the most appropriate predictor for survival in the subgroups of patients with malignant neoplasm and chronic cardiac disease” is not supported by the data presented. The publication has several biases, deficiencies and errors. First, the statement that “all non-survivors displayed a strong selenium deficit ([Se] < 55.2 µg L^−1^) already at hospital admission” is denied by Figure 2, which shows that at least two of the patients who died at the end of the study had, at admission, a selenium concentration over this amount. Similarly, the claim that “none of the patients were still presenting a Se status below 55.2” is contradicted by Figure 3A (see the original article). The length of hospital stay is considered as a factor in the discussion; however, data is lacking and reported only for a few extreme cases. Finally, the selenium thresholds are unclear and no indication is given regarding the vaccination status (based on the Journal’s info, the publication was submitted at the end of August 2021 and ethics was approved at the end of March 2021, i.e., after vaccines became available).

In November 2021, Skesters and coworkers submitted a work that was published in February 2022; it reported significantly lower blood selenium levels in 40 “acutely ill” COVID-19 patients, half of which were “treated in the COVID-19 unit” and the other half of which were “treated in the intensive care unit” of the Pauls Stradiņš Clinical University Hospital of Riga (Latvia), compared to 80 “post-COVID-19 disease patients”, half of which were “infected with COVID-19 in the spring–early summer period (1st wave)” and the other half of which were “half summer–autumn patients (2nd wave)” [70]. The work has several shortcomings and errors. The selenium levels of patients “treated in the COVID-19 unit” and the disease severity of post-COVID-19 patients are missing, the discharge time of the post-COVID-19 patients is unclear, and no indication is given as to whether the patients were vaccinated or not.

Chanihoon et al. compared the selenium levels in different biological samples (scalp hair, blood, serum, nasal fluid, sputum and saliva), obtained between October 2021 and January 2022 from multiple hospitals in Sindh (Pakistan), of smoking and non-smoking COVID-19 patients (*n* = 52 + 63) and healthy referents (*n* = 71 + 87) [71]. Based on their data, the selenium levels of COVID-19 patients (categorized into mild/severe/critical) were about halved in all biological samples, compared to respective smokers and non-smokers. However, even if a statistical analysis is mentioned, the association between selenium levels and COVID-19 severity was not analyzed. In addition, the patients’ vaccination status was missing, numerical inconsistencies were present and, worst of all, the COVID-19 patients were also smear-positive for pulmonary tuberculosis.

**Table 5 molecules-28-04130-t005:** Observational studies reporting association between selenium and COVID-19.

First Author, Year [Ref]	Country	Study Period	N of Subjects Involved	Study Population (Sex, Mean Age)	Selenium Plasma Levels	Follow Up
Zhou, 2022 [53]	China	27 January–18 May 2020	16	Distribution according to COVID-19 disease severity: mild group males *n* = 5, females *n* = 3 (mean age 42.5 ± 6.93 years); severe group males *n* = 4, females *n* = 4 (mean age 51.9 ± 16.7 years)	Mild about 125 µg/Kg, severe about 90 µg/Kg	No
Im, 2020 [54]	South Corea	February–June 2020	200	Males *n* = 29, females *n* = 21; mean age 52.2 ± 20.7	Median value Females 96.7 ng/mL (90.6–107.8); males 101.4 ng/mL (86.9–105.7); total 98.3 ng/mL (90.3–107.6)	Yes
Skalny, 2021 [55]	Russia, Norway, France, Greece	March–June 2020	194	Distribution according to COVID-19 disease severity: mild *n* = 50 (mean age 50.47 ± 15.91), moderate *n* = 50 (mean age 54.22 ± 12.5), severe *n* = 50 (mean age 64.5 ± 15.49); control group *n* = 44 (mean age 55.67 ± 4.36); Gender distribution (males/females): control 7/16; mild 25/25; moderate 31/19; severe 25/25	Control 0.102 ± 0.016 μg/mL, mild 0.093 ± 0.020 μg/mL moderate 0.090 ± 0.022 μg/mL, severe 0.087 ± 0.031 μg/mL	No
Pincemail, 2021 [56]	Belgium	May 2020	9	Mean age 64 (53–71); males *n* = 8, females *n* = 1	Reference Interval 73–110 µg/L, median (range) 74 µg/L (59–103)	Yes
Zeng, 2021 [57]	China	Not specified, submission date June 2020	138	Males *n* = 79, females *n* = 59 (total mean age 61.5); distribution according to COVID-19 disease severity: non-severe males *n* = 38, females *n* = 32 (global mean age 60); severe males *n* = 38, females *n* = 32 (global mean age 65)	Median value: all patients 22.95 μg/L (14.44–36.08) non-severe 25.55 μg/L (19.04–37.64) severe 20.27 μg/L (13.53–35.34)	Yes
Al-Saleh, 2022 [58]	Saudi Arabia, Canada	3 June–11 July 2020	155	Distribution according to COVID-19 disease severity: asymptomatic *n* = 16, mild *n* = 49, moderate *n* = 68, severe *n* = 22 age range: 18–95 years mean age 50 years. Females *n* = 78, males *n* = 77.	Mean ± SD: asymptomatic 86.56 ± 18.95 μg/L, mild 78.36 ± 18.04 μg/L, moderate 87.51 ± 19.26 μg/L, severe 76.6 ± 23.54 μg/L. Total 82.97 ± 19.91 μg/L	No
Moghaddam, 2020 [59]	Germany	Not specified, submission date July 2020	33	Females *n* = 19, males *n* = 14 (mean age 77 years). Death: females *n* = 4, males *n* = 2 (mean age 89 years); discharge: females *n* = 15, males *n* = 12 (mean age 69 years). Reference cohort of EPIC (European Prospective Investigation into Cancer and Nutrition)	Mean ± SD: all Samples 50.8 ± 15.7 µg/L, discharge 53.3 ± 16.2 µg/L, death 40.8 ± 8.1 µg/L	No
Heller, 2021 [60]	Germany	Not specified, submission date August 2020	35	Females *n* = 19, males *n* = 14 (mean age 77 years). Death: females *n* = 4, males *n* = 2 (mean age 89 years); discharge: females *n* = 15, males *n* = 14 (mean age 70 years). Reference cohort of EPIC (European Prospective Investigation into Cancer and Nutrition)	No data	No
Hackler, 2021 [61]	Germany	Not specified, submission date April 2021	35	Females *n* = 20, males *n* = 15 (mean age 77 years). Death: females *n* = 5, males *n* = 2 (mean age 89 years); discharge: females *n* = 15, males *n* = 13 (mean age 69 years). Reference cohort of EPIC (European Prospective Investigation into Cancer and Nutrition)	No data	No
Soto, 2022 [62]	Mexico	20 August–20 September 2020	86	COVID-19 patients *n* = 61; males *n* = 44, females *n* = 17 (global mean age 56 ± 13 years); distribution according to COVID-19 disease severity: moderate *n* = 34 (mean age 54 ± 12), (females *n* = 23, males *n* = 11); severe *n* = 27 (mean age 59 ± 14 years), (females *n* = 6, males *n* = 21)	No data	No
Erol, 2021 [63]	Turkey	15 July–15 December 2020	141	Control pregnant patients *n* = 70, COVID-19 pregnant patients *n* = 71 in different trimesters; I trimester: control patients *n* = 26 (mean age 26.34 ± 4.02 years), COVID-19 patients *n* = 24 (mean age 28.37 ± 4.70 years); II trimester: control patients *n* = 22 (mean 28.0 ± 6.27 years), COVID-19 patients *n* = 26 (mean age 29.76 ± 6.66 years); III trimester control patients *n* = 22 (mean age 26.3 ± 4.11 years), COVID-19 patients *n* = 21 (mean 28.95 ± 4.77 years).	No data	No
Majeed, 2021 [64]	India	Not specified, submission date September 2020	60	Control patients *n* = 30 (males *n* = 14, females *n* = 16) (mean age 33.5 years); COVID-19 patients *n* = 30 (males *n* = 24, females *n* = 6) (mean age 40.5 years)	Mean ± SD: COVID patients 69.2 ± 8.7 ng/mL, controls 79.1 ± 10.9 ng/mL	No
Younesian, 2022 [65]	Iran	Not specified, submission date March 2021	100	Control patients *n* = 50; COVID-19 patients *n* = 50 (*n* = 13 non-survivor group–mean age 72 years; *n* = 37 survivor group - mean age 49 years)	Mean ± SD: COVID-19 patients 77. 8 ± 13.9 μg/L, healthy control individuals 91.7 ± 16.7 μg/L, females COVID-19 patients 77.8 ± 16.7μg/L, healthy control individuals 95.8 ± 18.8 μg/L, males COVID-19 patients 77.8 ± 12 μg/L, healthy control individuals 88.4 ± 14.5 μg/L, survivor group 77.9 ± 14.3 μg/L, non-survivor group 77.2 ± 12.3 μg/L	No
Kirankaya, 2022 [66]	Turkey	December 2020–May 2021	195	COVID-19 patients *n* = 146 (males *n* = 64, females *n* = 82; mean age 120 months). Control group *n* = 49 (males *n* = 26, females *n* = 23; mean age 70 months). COVID-19 hospitalized *n* = 38 (males *n* = 19, females *n* = 19; mean age 87 months); COVID-19 outpatients *n* = 108 (males *n* = 45, females *n* = 63; mean age 124 moths)	Mean ± SD: control group 66.5 + 11.4 μg/L, COVID-19 (+) outpatients 58.8 + 8.3 μg/L, COVID-19 (+) hospitalized 52.1 + 9.6 μg/L	No
Kocak, 2022 [67]	Turkey	Not specified, submission date July 2021	92	COVID-19 patients *n* = 60 (males *n* = 32, females *n* = 28; mean age years 48.8 years); distribution according to COVID-19 disease severity: asymptomatic *n* = 4 (males N2=, females *n* = 2; mean age years 41.25 years); mild *n* = 15 (males *n* = 13, females *n* = 2; mean age 31.9 years); moderate *n* = 28 (males *n* = 13, females *n* = 15; mean age 54.21 years); severe *n* = 13 (males *n* = 4, females *n* = 9; mean age 58 years). Control group *n* = 32 (males *n* = 11, females *n* = 21; mean age years 45.5 years)	Se (ppb) control patients 255.23 ± 42.67 μg/kg, COVID-19 patients 215.34 ± 49.83 μg/kg; asymptomatic 236.17 ± 52.82 μg/kg, mild 196.85 ± 41.04 μg/kg, moderate 226.15 ± 48.79 μg/kg, severe 206.97 ± 57.18 μg/kg.	No
Ozdemir, 2022 [68]	Turkey	January–August 2021	15	Moderate COVID-19 patients *n* = 15 (males *n* = 12, females *n* = 3; mean age 58.93 ± 6.70 years)	Before medical treatment 71.51 µg/L (65.08–86.69); after medical treatment 88.14 µg/L (79.08–109.85)	Yes
Du Laing, 2021 [69]	Belgium	Not specified, submission date August 2021	138	COVID-19 patients *n* = 138 (study 1 on plasma: *n* = 79 patients; study 2 on serum: *n* = 59	Mean value 56.6 µg/L; males 55, 57.6 µg/L, females 24, 54.2 µg/L, until 65 years 58.2 µg/L; above 65 years 53.9 µg/L, malignant neoplasm 50.2 µg/L, malignant neoplasm 66, 57.8 µg/L; diabetes 52.2 µg/L, diabetes 58.5 µg/L; obesity 52.2 µg/L, obesity 48 µg/L; chronic cardiac disease 57.0 µg/L	Yes
Skesters, 2022 [70]	Uk, Lettonia	Not specified, submission date November 2021	120	80 post-COVID-19 disease patients and 40 acutely ill patients. Other data non-specified	The extreme limits (min/max) were from 75.4 µg/L to 43.2 µg/L, acute disease 69.7 µg/L, no COVID Spring–summer wave 84.6 µg/L, no COVID Summer–autumn wave 88.2 µg/L	No
Chanihoon, 2022 [71]	Pakistan, UK, China	October 2021–January 2022	115	COVID-19 patients *n* = 115 (non-smokers *n* = 63; smokers *n* = 52); control group *n* = 43 (non-smokers *n* = 19; smokers *n* = 24); 29–59 years. Distribution according to COVID-19 disease severity: non-smokers, mild *n* = 49, severe *n* = 12, critical=2; smokers, mild *n* = 52, severe *n* = 46, critical *n* = 2	Referents non-smokers 232 ± 15.9 μg/L, smokers 209 ± 12.0 μg/L, COVID-19 patients non-smokers 125 ± 9.97 μg/L, smokers 102 ± 8.95 μg/L	No

### 3.3. Reviews on Selenium and COVID-19

As mentioned in the introduction, immediately after the beginning of the COVID-19 pandemic, several reviews speculated on the association between COVID-19 and nutrients. With respect to selenium, the most reported topic concerned the effect of selenium deficiency on the virulence of Coxsackievirus B3 and influenza virus virulence. We have analyzed the contents of some of these reviews and of others that are highly cited.

#### 3.3.1. Reviews on Virulence in Selenium-Deficient Mice

COVID-19 reviews that report results from the group of Dr. Beck, which attend to the effects of dietary selenium deficiency on coxsackievirus B3 [17,72,73,74] or influenza virus [75,76,77] virulence in mice, deserve a separate paragraph. In fact, among the 211 reviews here found (Figure 1), 85 cite one or more of Beck’s articles. Notably, one earlier recapitulative publication from Beck’s group [72] has been cited in 15 reviews or books since the beginning of the pandemic [78,79,80,81,82,83,84,85,86,87,88,89,90,91,92], including a widely read one [78] that is itself cited by 790 articles, based on Scopus.com. Unfortunately, this recapitulative paper contains inconclusive and contradictory results, and was indeed subsequently revised by the same group. Firstly, no indication is given as to how much selenium was administered with the diet, and the selenium status was analyzed indirectly using serum glutathione peroxidase as a biomarker (dropping from 33.0 ± 4.4 to 4.7 ± 0.2 munits/mg protein [17,74]). Secondly, in virus-susceptible mice, vitamin E deficiency was associated with the same cardio-histopathologic effects as selenium deficiency, but both deficiencies were necessary in the virus-resistant mice [93]; this suggests that the cardio-histopathologic effects were rather due to an increased oxidative status, as suggested by lower serum glutathione peroxidase activity. Accordingly, selenium deficiency was not associated with lower natural killer cell activity nor with lower neutralizing antibody titers, indicating that, even with high viral loads, the immunological functions were mostly preserved in selenium-deficient mice (as stated by the authors themselves [17]), as shown also by normal viral clearance [74]; this indicates that other factors underlie the coxsackievirus B3 epistatic phenomena (see below) and its ability to induce mice cardiomyopathy.

In later studies using influenza virus, Beck and coworkers quantified the selenium levels in diets, livers (dropping from 687 ± 41 to 81 ± 16 µg/kg) and sera (dropping from 489 ± 55 to 44 ± 15 µg/L) [75,76,77]. However, contradictory results were reported. In fact, the inoculation of selenium-deficient mice with a mild strain of influenza was associated with an increased inflammation of the lung (graded semi-quantitatively by eye inspection) and an increased number of infiltrating bronchoalveolar cells (by FACS analysis); however, it was not, unlike coxsackievirus B3 infection, associated with higher viral titers (both with mild [75] and virulent [76] influenza strains) compared to control mice fed with selenium. Consistently, no reduction in immunological functions was observed (antibodies and viral clearance). On the other hand, consistent with the inflammatory phenotype, the expression of genes coding cytokines and chemokines in mediastinal lymph nodes was altered in selenium-deficient mice, compared to the control-fed ones. However, no analysis of lung inflammation or bronchoalveolar were performed, nor was the expression of genes analyzed in mock-infected, selenium-deficient, mice, which is one important bias. In fact, the absence of such analysis cannot exclude the presence of lung inflammation before virus inoculation. Accordingly, the authors found the early upregulation of genes coding for chemokines and the downregulation of those coding for anti-inflammatory cytokines; they also found γ-IFN that were reduced at all time points [75], indicating compromised immune functions. Thus, even if such studies do not disprove that selenium deficiency can influence the pathogenicity of a virus, they surely do not demonstrate that selenium supplementation in healthy subjects can restore impaired immune responses to viruses, at least not better than could be obtained by other supplementations, such as vitamin E; this is demonstrated by the fact that vitamin E deficiency [72] or a genetically induced lack of GPX activity in KO mice [75] yielded a phenotype that was identical to that yielded by selenium deficiency. As hypothesized by the same authors, selenium deficiency could induce an increase in oxidative stress [75], actually non-unexpectedly because of the observed huge drop in liver and serum. Accordingly, except for the decreased glutathione peroxidase activity, such a hypothesis was verified 6 years later when Beck and coworkers reported lower levels of liver and lung TR in mice prior to inoculation [94], slight but statistically significant lower levels of liver GS and GSH [76,94], increased levels of liver and lung SOD, some degree of lung pathology [94], but similar levels of all other markers of inflammation tested at day 0 [76].

In the already mentioned highly cited review by Zhang and Liu [78], the authors slavishly copied from Guillin et al. [95] the hypothesis that “dietary selenium deficiency that causes oxidative stress in the host can alter a viral genome so that a normally benign or mildly pathogenic virus can become highly virulent in the deficient host under oxidative stress”. It should be remarked that the mentioned epistatic phenomenon was reported only in mice by Beck and coworkers in 1995 for coxsackievirus [73] and in 2001 for mild, non-host adapted influenza virus [75,77]; this is however, to our knowledge, still lacking elucidation. Indeed, the hypothesis that increased oxidative stress could induce mutations in the viral genome through direct oxidative damage to the viral RNA or could induce the selection of the new consensus sequence following the duplications induced by the error prone RNA-dependent polymerase [77] still awaits demonstration. It is noteworthy that, in the last publication from Beck and coworkers, the strong reduction in GPX activity in selenium-deficient mice only slightly affected liver GS and GSH levels, and did not induce viral genome mutations nor the increased virulence of an host-adapted influenza virus [76]; this demonstrates that selenium deficiency is not determinant for virus pathogenicity nor lethality but that, on the contrary, selenium deficiency significantly increases the resistance of mice to influenza [76], and is not sufficient for the induction of epistatic phenomena.

Aside from Beck’s work on influenza and coxsackievirus discussed above, the review by Zhang and Liu [78] reports the other self-styled skills of selenium. For example, in their highly cited review, one can read that selenium deficiency “induces rapid mutation of benign variants of RNA viruses to virulence”, a sentence whose reference is another review, that by Harthill [96]. Such an assumption is unsupported. In fact, it comes from the observation of additional temporal temperature gradient electrophoresis (TTGE) bands in the PCR products of poliovirus obtained from the feces of subjects taking the oral live attenuated poliomyelitis vaccine but that are not supplemented with selenium, compared to subjects receiving a supplementation of selenium [97]. However, no sequencing of PCR products was performed in this publication and the reliability of the qualitative TTGE-reported results is questionable. Specifically, TTGE was apparently performed using PCR products with significant differences in their quantities, clearly lower and barely amplifiable in non-supplemented subjects, as shown in Figure 4 of the original work [97]. Besides the fact that the PCR approach was not quantitative (which could affect the reliability of the result on the greater clearance of the virus in the supplemented subjects), such huge differences in amplicon amounts can influence the melting temperature of their domains, affecting amplicon TTGE gel mobility, in addition to their specific sequence. Thus, the conclusion that host selenium supplementation reduces poliovirus mutations [97] appears not sufficiently substantiated.

#### 3.3.2. Other Reviews

One of the most cited reviews is that by Jayawardena and coworkers entitled “Enhancing immunity in viral infections, with special emphasis on COVID-19: A review” [98]. Based on Scopus.com, at the time of the submission of this work, Jayawardena’s review had 311 citations, including 129 reviews. The review recapitulates studies supplementing selenium alone or in multi-nutrient supplements (a total of 8 studies for minerals), and concludes that a zinc plus selenium daily supplementation (150 mg and 200 mg/daily respectively) “could be beneficial to improve immunity during viral infections” [98]. The following are some of the issues we found. Firstly, in one of the randomized controlled trials reviewed, individuals with normal baseline levels of selenium were inexplicably included in the low-selenium control group [99]. Secondly, in another randomized controlled trial reviewed, even if selenium and zinc supplementation increased humoral response and reduced the incidence of respiratory tract infections (for the 2 years of the study) following the influenza vaccination of elderly subjects, no significant differences were found in the survival rates [100]. It is noteworthy that, as reported in Jayawardena’s review itself, a daily supplementation of selenium in a multivitamin–mineral capsule did not ameliorate the severity of acute respiratory tract infections in well-nourished elderly individuals, nor did it change the incidence of acute respiratory tract infections during the median observation period of 441 days [101]. It is worth mentioning that the composition of the capsule was based on a paper that was later retracted [102].

Another highly cited review is that of Iddir et al. entitled “Strengthening the Immune System and Reducing Inflammation and Oxidative Stress through Diet and Nutrition: Considerations during the COVID-19 Crisis” [103]. The review is full of poorly founded evidence and parallels, including selenium geographical variabilities [104] and parallel roles in the hosts’ antioxidant defense system [105], ending with unsupported supplementation suggestions that are aimed at increasing the immune response [106] or improving of the vaccine response [107].

The review untitled “Selenium supplementation in the prevention of coronavirus infections (COVID-19)” [108] is an example of information distortion. In fact, the review attributes the statement “selenium supplementation inhibited the development of polio and influenza virus” to the already mentioned review of Jayawardena et al. [98], presumably referring to two different studies: (1) reference 33, a study performed using not only selenium but also zinc, and only reporting a higher production of post-influenza vaccine antibodies; (2) reference 40, exclusively reporting increased polio clearance based on a non-quantitative PCR of the viral gene in feces. Moreover, the authors state that selenium (selenite) “prevents virus entering to the healthy cell cytoplasm” thanks to an undemonstrated loss of the spike protein’s ability “to undergo the exchange reaction with disulfide groups of cell membrane proteins”, mediated by an undefined protein disulfide isomerase.

On the other hand, the review by Shakoor and coworkers bodes well because it starts with a question mark: “Immune-boosting role of vitamins D, C, E, zinc, selenium and omega-3 fatty acids: Could they help against COVID-19?” [109]. However, even if the authors rightly write that there is a lack of evidence (“clinical trials based on the associations of diet and COVID-19 are lacking”), they end the article with the following scientifically unsupported statement: with “the negligible risk profile of supervised nutritional supplementation, weighed against the known and possible benefits, it appears pertinent to ensure adequate, if not elevated intake of these key vitamins and minerals in people both at risk of, and suffering from COVID-19”. Fortunately, in this case, the review is hardly mentioned, with just one citation from the same group of authors [110].

## 4. Discussion

All reports, including those concluding that there is a lack of an association between selenium and COVID-19, have some limitations, which, however, are partially justified by the difficulty of conducting studies during the pandemic. Specifically, the estimation of sample numerosity, in order to determine the statistical power of results prior to the onset of the studies, was performed only in one study [38] among the 26 clinical observational and the 2 clinical supplementation studies, and the inhomogeneous distribution of the participants to the studies (sample size, mean ± sd: 115.2 ± 86.7, ranging from 15 to 367) makes it virtually impossible to compare the studies numerically. In addition to small n-numbers, the only two clinical supplementation studies here considered (among the five available to date) are both biased by the inhomogeneous administration of non-specific (also containing zinc) and non-personalized (without considering the individual status) supplements among patients. Other shortcomings of clinical supplementation studies include the lack of patient follow up (79%), the complete inhomogeneity of food and supplement intake before and during the hospitalization period, and the total lack of patient vaccination status (for studies performed after vaccines became available). Moreover, the data were incomplete for SARS-CoV-2 positivity (31%), selenium status (15%) and co-morbidities (58%). In the absence of indications of this kind and the dosage of supplements, no clear conclusions can be made regarding the potential role of selenium supplementation in COVID-19 severity or outcome, nor if the decline in serum selenium levels is a cause or rather a consequence of COVID-19 severity.

The use of genetically determined selenium levels can, in turn, eliminate the issues related to the declining circulating selenium levels linked to COVID-19, theoretically allowing selenium levels to be causally attributed to COVID-19 pathogenesis. The three genetic studies conclude that there is poor evidence regarding the association between selenium and COVID-19 outcomes. However, this approach has the limitation of not accurately predicting micronutrient concentrations, thus misjudging those that may also be most important in the patients most vulnerable to COVID-19 fatality. The available studies suffer a total lack of the direct measurement of the selenium status (a partial measurement of nail selenium was performed in only one out of three studies conducted).

With respect to the ecological studies, they mainly all suffer due to their dated (dating from 1985 or 2011 or 2017) and incomplete data (some cities are lacking or were arbitrarily omitted), the fact that confounders were not evaluated (age, chronic diseases and therapies), and regarding their indirect selenium status (intake inferred from non-specific food items or hair concentration). In fact, because of individual eating habits, the selenium status in the population does not necessarily correlate with the selenium concentration in crops or topsoil [111], and no studies highlighting the correlations between serum selenium levels and local crops or topsoil have been made available in the cited literary studies. Moreover, lockdowns and the conditions of fear linked to the rapid diffusion of the virus have further complicated “normal” dietary habits, which already suffer from migration biases that are typical of ecological studies. However, even if the hypothesis of a positive role of supplementation is not substantiated by ecological studies, the hypothesis that selenium status can be affected by its concentration in the soil is interesting; therefore, further research that tests COVID-19 incidence, for example in the northern part of India, where soil selenium levels are normal, would be useful in order to substantiate the role of this mineral in COVID-19 susceptibility.

## 5. Conclusions

Scientific reports published during the COVID-19 pandemic do not support selenium supplementation, neither for preventing SARS-CoV-2 infection nor for ameliorating COVID-19 prognosis, and particular attention should be given also to the reviews that came out during the first months of the COVID-19 pandemic. Further studies are needed to verify the benefits of selenium, in any, in reducing the risk or severity of COVID-19. These should preferably be chronic clinical supplementation studies in large cohorts, whose sample numerosity should be calculated in advance in order to set the statistical power of the results; this should be performed through the application of statistical formulas that are correlated to the COVID-19 parameters of interest. Hospitalized patients should be rapidly screened and recruited according to equivalent accurate criteria (age, sex, physical characteristics, SARS-CoV-2 positivity, selenium deficiency, symptoms severity) and comorbidities should be avoided, unless they are numerically consistent. Indeed, as international guidelines on critically ill patients currently recommend the use of supplements only in deficient patients [112], because of non-definitive evidence [113], patients with strong selenium deficiency should be avoided. In fact, independent of COVID-19, mineral supplementation is known to be beneficial in patients who are deficient, thus one should not be surprised if non-deficient patients show a lower risk of death. All patients should follow a specific and controlled diet, treated with the same therapeutic regimens (antibiotics, antivirals), and supplemented with a specified amount of selenium. Levels of plasma selenium should be measured at the baseline, prior to the start of the intervention phase, and at defined interval times during the supplementation, in concomitance with the other established parameters, including the viral load. A control group with similar characteristics should follow parallelly the identical study protocol, but without selenium supplementation. Due to the inability to follow them correctly, outpatients should be avoided as controls. Apart from the prognosis, other parameters should include selenium-related trace elements, antioxidants and biomarkers of oxidative damage.

## Figures and Tables

**Figure 1 molecules-28-04130-f001:**
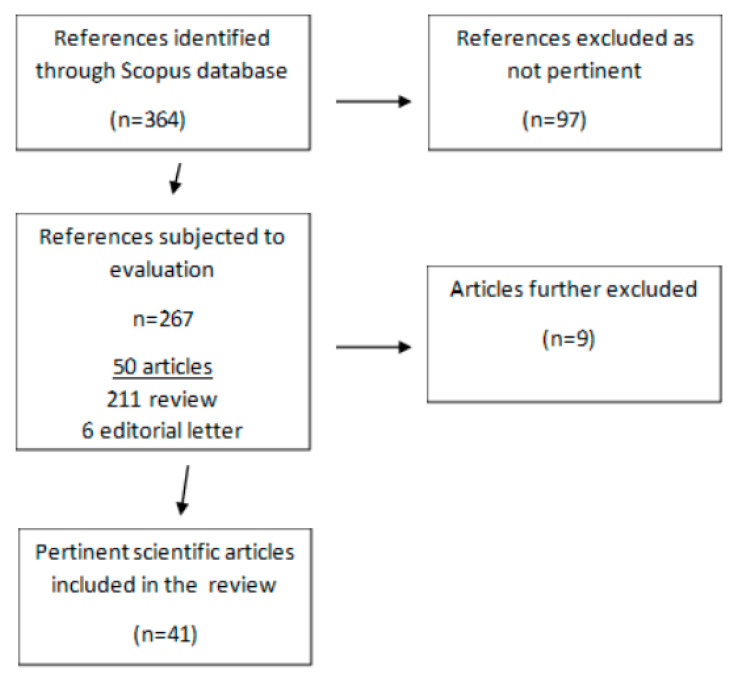
Flowchart of the selection process for the scoping review.

**Figure 2 molecules-28-04130-f002:**
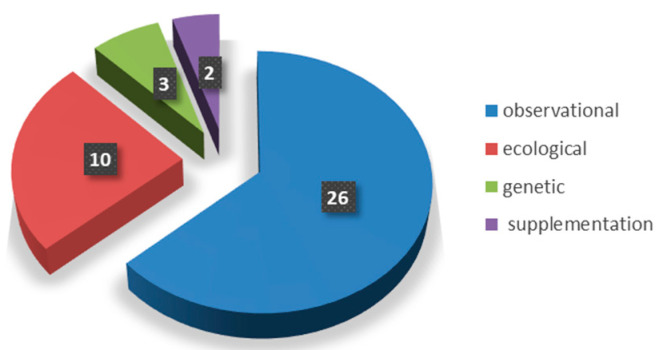
Characteristics of COVID-19 publications included in the review.

**Table 1 molecules-28-04130-t001:** Ecological studies reporting NO association between selenium and COVID-19.

First Author, Year [Ref]	Country	Study Period	Consulted Databases for Selenium Levels	Consulted Databases for COVID-19 Incidence
Galmés, 2020 [27]	Spain, Belgium, Italy, UK, Portugal, France, the Netherlands, Germany, Denmark, Finland	May 2020	Essential nutrients intake for the maintenance of the immune system, endorsed by the European Food Safety Authority	Worldometers.info COVID-19 epidemiological indicators
Galmés, 2022 [28]	Spain	Not specified	Micronutrient intake in Spain obtained from the household Spanish report, containing pre-pandemic (2019) consumption data (Ministry of Agriculture, Fisheries and Food)	Epidemiological COVID-19 Spanish data from the Centre for the Coordination of Health Alerts and Emergencies of the Ministry of Health (Update No. 235 of the Coronavirus Disease (COVID-19) on 23 October 2020)
Ermakov, 2022 [29]	Russia	September 2020–January 2021	Selenium concentration in herbaceous plants (cuts), surfaces and groundwaters, annual precipitations, and incidence of white muscle disease in farm livestock	Coronavirus situations in Russia on 29 January 2021, and on 5 September 2020, from www.koronavirustoday.ru/news/russia.
Chen, 2022 [30]	USA	8 October 2020–25 March 2021	Geochemical concentrations of selenium from the National Geochemical Survey 1997–2009	Epidemiological data on the case fatality rate of COVID-19 in USA, accessed using the COVID-19 interactive map from the “John Hopkinson” University
Nimer, 2022 [31]	Jordan	March–July 2021	Selenium supplementation data collected via a self-administered questionnaire using a Google form	COVID-19 disease symptoms and hospitalization status data collected via a self-administered questionnaire using a Google form

**Table 2 molecules-28-04130-t002:** Observational studies reporting NO association between selenium and COVID-19.

First Author, Year [Ref]	Country	Study Period	N of Subjects Involved	Study Population (Sex, Mean Age)	Selenium Plasma Levels	Follow Up
Eden, 2021 [32]	United Kingdom	March–May 2020	72	Males *n* = 54 (mean age 57.1 ± 9.8 years), females *n* = 18 (mean age not specified); survived *n* = 48	Selenium measured in 33 of 72 (46%) patients. Mean levels 0.88 µmol/L	No
Voelkle, 2022 [33]	Switzerland	17 March–30 April 2020	57	Males *n* = 34, females *n* = 23, mean age 67 years	Mean levels 0.96 µmol/L	No
Fromonot, 2022 [34]	France	24 April–23 May 2020	240	COVID-19 patients *n* = 152, non-COVID-19 patients *n* = 88, mean age 65 years	No data	No
Karakaya, 2021 [35]	Turkey	15 May–15 June 2020	49	Females *n* = 27, males *n* = 22; 8–18 year	Mean levels 66.4 µg/L	No
Tayyem, 2021 [36]	Qatar and Jordan	17 March–25 July 2020	367	Males *n* = 242 (mean age 44.0 ± 14.7 years), Females *n* = 125 (mean age 39.0 ± 16.1 years)	No data	No
Bagher, 2021 [37]	Iran	10 October–10 December 2020	226	Males *n* = 112, females *n* = 114 (mean age 56.36 ± 18.54 years); distribution according to COVID-19 disease severity: severity group males *n* = 56, females *n* = 56; non-severity group males *n* = 58, females *n* = 56	126.61 ± 2.05 μg/L all patients; 130.19 ± 3.19 μg/L severe group; 123.06 ± 2.58 μg/L not severe group, recovered patients 125.77 ± 2.41 μg/L deceased patients 129.15 ± 3.91 μg/L; <55 years 128.92 ± 34.01 μg/L; ≥55 years 124.48 ± 27.37 μg/L; Male 128.40 ± 31.29 μg/L; Female 124.80 ± 30.23 μg/L	No
Razeghi, 2021 [38]	Iran	Up to 1 September 2020	84	Females *n* = 37, males *n* = 47; distribution according to COVID-19 disease severity: mild *n* = 38 (females *n* = 15, males *n* = 23); moderate *n* = 27 (females *n* = 15, males *n* = 12); severe *n* = 19 (female *n* = 7 male *n* = 12); mean age 64.66 ± 11 years (mild 51 ± 14; moderate 59 ± 14; severe 81 ± 7)	Mild group 47.07 ± 20.82 ng/mL, moderate group 47.36 ± 25.6 ng/mL, severe group 29.86 ± 11.48 ng/mL	No

## Data Availability

The methodology, and all the elaboration generated and used for this review, are fully available upon request from the corresponding author.

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
