# Peer review of "Could Selenium Supplementation Prevent COVID-19? A Comprehensive Review of Available Studies"

_molecules, 2023, doi:10.3390/molecules28104130_

Round 1

Reviewer 1 Report

It is quite interesting to read a review article, which is focused on demystifying an agent(s) suggested in combat against a certain disease. The authors have given a very stern and critical review of published articles regarding the selenium use in COVID.

I believe that the paper is interesting, however, bearing in mind that that quite a long period has passed since the appearance of novel disease (I believe it cannot even be called novel, for that matter)- we can safely say that not one single thing will affect the course or outcome of the disease, especially Se status, rather than numerous factors.

I invite authors to add a paragraph, describing, in details, a study which would be a perfect one for assessing Se influence in COVID. Type of study, number of participants, all about biochemical or other studies... The idea of review articles is not only to give the overview of previously published paper, but to give idea for new researchers or study groups which plan to investigate the topic reviewed. 

Additional comment- please avoid starting the sentences with number, eg. 15 studies conclude for 64 absence of significant association,.....

Author Response

We thank the Reviewer for the observation about the need of including a description on how a study should be performed for assessing Selenium influence in COVID-19. The point was addressed in the Conclusion paragraph.

All numbers that were at the beginning of sentences have been converted to letters.

Reviewer 2 Report

The authors have made an interesting comprehensive review of the  precision and reliability of data from scientific papers on selenium status, supplementation, and COVID-19. Overall, the paper is well written, and it is of interest to molecules reader. However, some changes should be taken into consideration before being accepted for publication.

Line 2: title is too long, I suggest deleting the expression “of supplementation, genetic, ecological, and observational studies”.

Line 45: please keep the headline of this part as “methodology”.

Line 112 please specify if it was selenium or other minerals supplementation.

I recommend using the term "clinical observational/supplementation studies" in the manuscript rather than "observational/supplementation studies".

Line 154 The abbreviation "DMGDH" is not explained in the manuscript.

Line 169: The abbreviation “GWAS” is not explained in the manuscript.

Line 177: headline to be checked for English errors and reformulated (too long headline)

Section 3.2.3 This part is too verbose and should be shortened and to be more concise

Section 3.3.1 It is not clear why the authors chose Coxsackievirus B3 and influenza virus to be linked to COVID-19 and selenium status. The author should add in the text separately the scientific argument of choosing these two viruses in comparison to SARS-COV2, the virus responsible for COVID-19.

Line 675 I recommend making this part as a conclusion not as a discussion. Discussion is already made by authors in each previous part.

Author Response

Please find below the point-by-point reply to reviewer’s comments:

Line 2: title is too long, I suggest deleting the expression “of supplementation, genetic, ecological, and observational studies”.

The title has been corrected as suggested: "Could Selenium supplementation prevent COVID-19? A comprehensive review of available studies

Line 45: please keep the headline of this part as “methodology”.

Done (line 48).

Line 112 please specify if it was selenium or other minerals supplementation.

Done (line 113).

I recommend using the term "clinical observational/supplementation studies" in the manuscript rather than "observational/supplementation studies".

Done throughout the manuscript.

Line 154 The abbreviation "DMGDH" is not explained in the manuscript.

Done.

Line 169: The abbreviation “GWAS” is not explained in the manuscript.

Done.

Line 177: headline to be checked for English errors and reformulated (too long headline)

The headline was shortened (line 178).

Section 3.2.3 This part is too verbose and should be shortened and to be more concise

All paragraphs were shortened so as not to exceed 10 lines for all studies analyzed, except for Du Laing and Schomburg publications.

Section 3.3.1 It is not clear why the authors chose Coxsackievirus B3 and influenza virus to be linked to COVID-19 and selenium status. The author should add in the text separately the scientific argument of choosing these two viruses in comparison to SARS-COV2, the virus responsible for COVID-19.

The title of section 3.3.1 was changed and the point addressed in the paragraph was introduced more clearly (lines 482-484).

Line 675 I recommend making this part as a conclusion not as a discussion. Discussion is already made by authors in each previous part.

The paragraph was divided in the two paragraphs Discussion and Conclusion.

Author Response

16-17 "we conclude that selenium supplementation is unsupported neither for preventing SARS- 16 CoV-2 infection nor ameliorating COVID-19 prognosis. Please change this sentence to the following sentence. 724-726“scientific reports on selenium supplementation available to date do not support neither a specific role of selenium in COVID-19 severity, neither the role of its supplementation as a tool to prevent the disease onset, nor its etiology.”

Done.

18 In the keywords, write the term "prevention".

Done.

25 Add that "EFSA" stands for The European Food Safety Authority

Done (line 27).

Reviewer 4 Report

Could Selenium supplementation prevent COVID-19? A comprehensive review of supplementation, genetic, ecological, and observational studies

The present review systematically examines the scientific evidence investigating selenium relationship with COVID-19. The review consolidates the findings from available literatures demonstrating the supplementation, genetic, ecological, and observational studies. The introduction can be revised by adding a short description of the reported pharmacological properties of selenium in various disease conditions. The approach and the overall design of the review are good.

Author Response

A sentence with a very short short description of many known pharmacological properties of selenium was added, including two related references (lines 21-24)

Round 2

Reviewer 1 Report

 No further suggestions, kind regards to Authors.

Reviewer 2 Report

The authors of the manuscript have answered all my questions and suggestions for changes and have incorporated them into the manuscript. I recommend accepting the manuscript as it is.